# A SAR Dataset of Ship Detection for Deep Learning under Complex Backgrounds

**Yuanyuan Wang [1,2]** , **Chao Wang [1,2]** , **Hong Zhang [1,\*]** , **Yingbo Dong [1,2] and Sisi Wei [1,2]**

1  Key Laboratory of Digital Earth Science, Institute of Remote Sensing and Digital Earth, Chinese Academy of Sciences, Beijing 100094, China; wangyy2016@radi.ac.cn (Y.W.); wangchao@radi.ac.cn (C.W.); dongyb@radi.ac.cn (Y.D.); weiss@radi.ac.cn (S.W.)
2  University of Chinese Academy of Sciences, Beijing 100049, China
*  Correspondence: zhanghong@radi.ac.cn; Tel.: +86-10-8217-8186

**Abstract:** With the launch of space-borne satellites, more synthetic aperture radar (SAR) images are available than ever before, thus making dynamic ship monitoring possible. Object detectors in deep learning achieve top performance, benefitting from a free public dataset. Unfortunately, due to the lack of a large volume of labeled datasets, object detectors for SAR ship detection have developed slowly. To boost the development of object detectors in SAR images, a SAR dataset is constructed. This dataset labeled by SAR experts was created using 102 Chinese Gaofen-3 images and 108 Sentinel-1 images. It consists of 43,819 ship chips of 256 pixels in both range and azimuth. These ships mainly have distinct scales and backgrounds. Moreover, modified state-of-the-art object detectors from natural images are trained and can be used as baselines. Experimental results reveal that object detectors achieve higher mean average precision (mAP) on the test dataset and have high generalization performance on new SAR imagery without land-ocean segmentation, demonstrating the benefits of the dataset we constructed.

**Keywords:** ship detection; SAR dataset; object detectors; deep learning; complex backgrounds

## 1. Introduction

Ship detection is important for marine surveillance in areas such as illegal fishing, oil spill detection, marine traffic management, and maritime piracy [1–6]. Currently, images from optical and reflected infrared, hyperspectral, thermal infrared, and radar are used [3]. Unlike the first three image types, radar captures targets by actively sending microwaves, thus leading to continuous imaging [7,8]. Synthetic aperture radar (SAR), a type of radar, is the most suitable for ship detection because its resolution is constant even when distant from the observed targets [8]. In recent years, many satellites such as the Sentinel-1 from Europe's Copernicus program [9], have been providing numerous SAR images for various maritime applications, making dynamic ship detection possible. Therefore, it is necessary to automatically process them to meet the marine surveillance needs [10].

Traditional ship detection methods are mainly based on constant false alarm (CFAR) [1,2,4,6]. The premise of these approaches is land-ocean segmentation, thus limiting the speed required to acquire ships [2–4,6]. In addition, these methods are typically dependent on the statistical distribution of sea clutter [1,2,11], leading to poor robustness for new SAR imagery.

Since they are able to learn hierarchical representations of the targets of interest, object detectors have been adapted in deep-learning community for ship detection in SAR images [12–21]. The typical process is to modify the pretrained object detectors for ship detection, and then train them for detection using SAR chips. Even if they achieve promising detection accuracy, the results are mainly for test ship chips. When applying these models to new SAR imagery, land-ocean segmentation is still necessary. If

there is no land-ocean segmentation, many false positives occur. For example, [16,20] have at least 15% false positives, which seems too high to be practical in an operational system. This may be due to inadequate training dataset [16,20]. On one hand, the training dataset contains few complex ship chips. On the other hand, the number of datasets is limited compared to the computer vision field [14–16,20].

Since the volume of datasets is low, when training the object detectors for ship detection in SAR images, fine-tuning or transfer-learning is widely used [14–16,20]. The deficiency is that current models fail to exploit the SAR characteristics. This is because the datasets for the pretrained model are natural images from ImageNet [22] or COCO [23], which are quite different from SAR images [7,8]. Unlike natural images from cameras, SAR is a coherent imaging process, leading to inherent characteristics such as foreshortening, layover, and shadowing [8]. Apart from the imaging mechanisms, ships in SAR images appear differently with regard to size and background. For the former, the same ship in different resolutions varies widely, and ships of various shapes in the same resolution have different sizes. For the latter, targets on the ocean, near an island, or in a harbor that have similar backscattering mechanisms to ships produce high false-positive rates. Therefore, it is necessary to construct a dataset that is as complex as possible to improve the application of deep learning for ship detection in SAR images.

Based on the analysis above, a dataset is constructed first. Specifically, it contains 102 Gaofen-3 images [24] and 108 Sentinel-1 images [9] that were used to create 59,535 ships in 43,819 ship chips. They vary in terms of polarization, resolution, incidence angle, imaging mode, and background. The detailed data information will be presented in Section 2. This dataset will be available online on our website [25]. Second, modified object detectors are adapted for ship detection and can be baselines. Our contributions are as follows.

1.  A SAR ship detection dataset under complex backgrounds is constructed. This dataset can be the catalyst for the development of object detectors in SAR images without land-ocean segmentation, thus helping the dynamic monitoring of marine activities.
2.  Modified state-of-the-art object detectors, including Faster regions with convolutional neural networks (R-CNN) [26], single shot multiBox detector (SSD) [27], and RetinaNet [28] are adapted to ship detection and can be baselines.

This paper is organized as follows. Section 2 presents the details of the dataset. Section 3 contains the experiment results. Sections 4 and 5 present the discussion and conclusions, respectively.

## 2. The SAR Ship Dataset

### 2.1. The Original SAR Image Dataset

There are 102 Gaofen-3 images and 108 Sentinel-1 images that were used to construct the dataset. For Gaofen-3, the images have resolutions of 3 m, 5 m, 8 m, and 10 m with Ultrafine Strip-Map (UFS), Fine Strip-Map 1 (FSI), Full Polarization 1 (QPSI), Full Polarization 2 (QPSII), and Fine Strip-Map 2 (FSII) imaging modes, respectively. For Sentinel-1, the imaging modes are S3 Strip-Map (SM), S6 SM, and IW-mode. Details of these images, including resolution, incidence angle, and polarization are presented in Table 1. It is obvious that these images vary widely. The coverage of these images is shown in Figure 1. Since the revisiting time is 12 days, we only show one S3 SM Sentinel-1 in Figure 1b and one S6 SM Sentinel-1 in Figure 1c.

**Table 1.** Detailed information for original synthetic aperture radar (SAR) imagery.

| Sensor | Imaging Mode | Resolution Rg. × Az. (m) | Swath (km) | Incident Angle (°) | Polarization | Number of Images |
|---|---|---|---|---|---|---|
| GF-3 | UFS | 3 × 3 | 30 | 20~50 | Single | 12 |
| GF-3 | FS1 | 5 × 5 | 50 | 19~50 | Dual | 10 |
| GF-3 | QPSI | 8 × 8 | 30 | 20~41 | Full | 5 |
| GF-3 | FSII | 10 × 10 | 100 | 19~50 | Dual | 15 |
| GF-3 | QPSII | 25 × 25 | 40 | 20~38 | Full | 5 |
| Sentinel-1 | SM | 1.7 × 4.3 to 3.6 × 4.9 | 80 | 20~45 | Dual | 49 |
| Sentinel-1 [1] | IW | 20 × 22 | 250 | 29~46 | Dual | 10 |

[1] Only single cross-polarization SAR images are used to construct the dataset. Ultrafine Strip-Map (UFS), Fine Strip-Map 1 (FSI), Full Polarization 1 (QPSI), Full Polarization 2 (QPSII), S3 Strip-Map (SM), and Fine Strip-Map 2 (FSII) imaging modes, respectively.

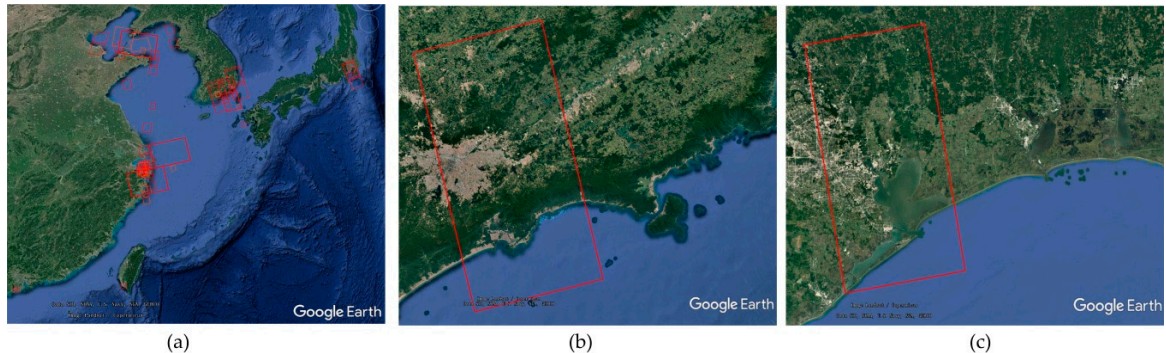

(a)    (b)    (c)

**Figure 1.** Image coverage of Gaofen-3 and Sentinel-1 IW (**a**), Sentinel-1 S6 SM (**b**) and Sentinel-1 S3 SM (**c**). The red rectangles indicate the coverage of each image.

Ships appear as bright pixels mainly due to the double reflection of radar pulses emitted by the sensor [5,6] as in Figure 2. However, due to the imaging conditions, the normalized measures of the pixel intensity, i.e., the calibrated backscatter coefficients (sigma0 values) vary with incidence angle and polarization. The incidence angle mainly influences the geometry of ships in SAR images, producing foreshortening, layover, and shadowing as indicated by red, magenta, and blue arrows in Figure 3, respectively. The geometry deformation poses challenges for ship detection. Generally, copolarization has higher sigma0 values than cross-polarization [8], as shown in Figure 4.

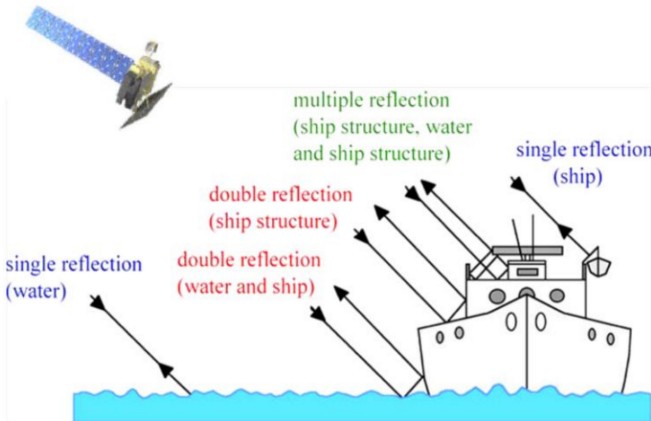

**Figure 2.** Scattering mechanisms of a ship and ocean in SAR images (calm ocean conditions) [6].

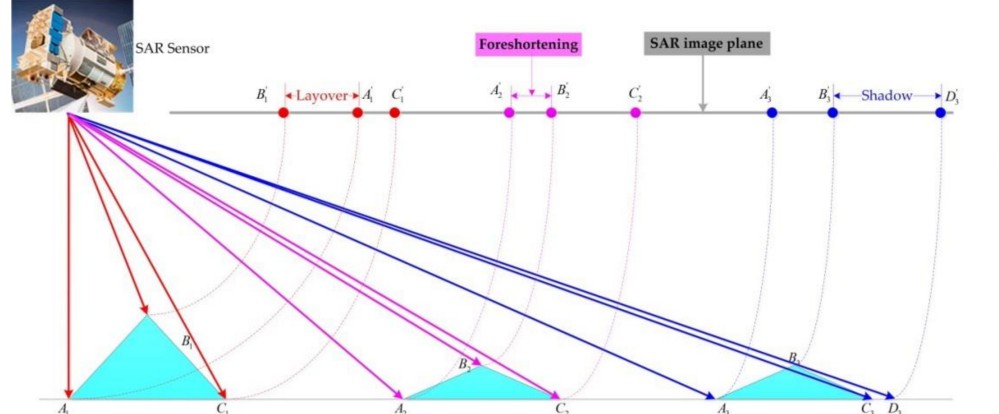

**Figure 3.** Illustration of the relationship between incidence angle and geometry deformation.

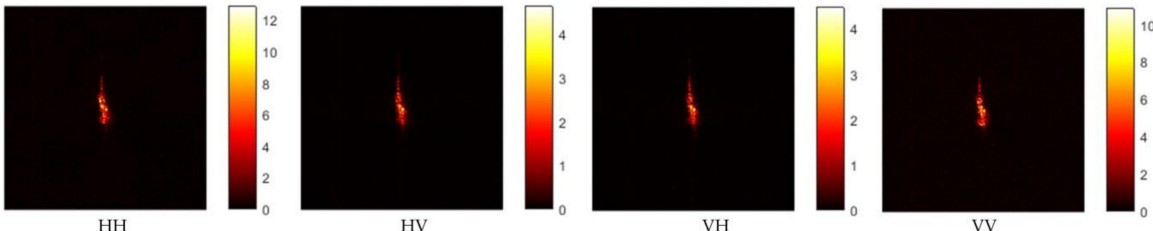

**Figure 4.** Illustration of the sigma0 values of the same ship under four polarization channels.

## 2.2. Policies for Construction of the Ship Detection Dataset

The procedure for the dataset construction is shown in Figure 5. First, all SAR images are converted to sigma0 values. Second, candidate subimages containing ships with sizes greater than 800 pixels in both range and azimuth are cropped. Third, sliding windows from these candidate subimages are used to acquire ship chips 256 × 256 pixels in size. To enrich the backgrounds of the ships, 128 pixels are shifted over both columns and lines during the sliding window, leading to 50% overlap of adjacent ship chips. Some examples of ship chips are shown in Figure 6. To mark the ship position, these ship chips are converted to gray images. Finally, they are labeled by SAR experts with LabelImg [29]. Each ship chip corresponds to an Extensible Markup Language (XML) file like that in the PASCAL VOC detection dataset [30], indicating the ship location, the ship chip name, and the image shape indicated by red, green, and cyan rectangles in Figure 7, respectively. Finally, the entire dataset is randomly split into a training dataset (70%), a validation dataset (20%), and a test dataset (10%).

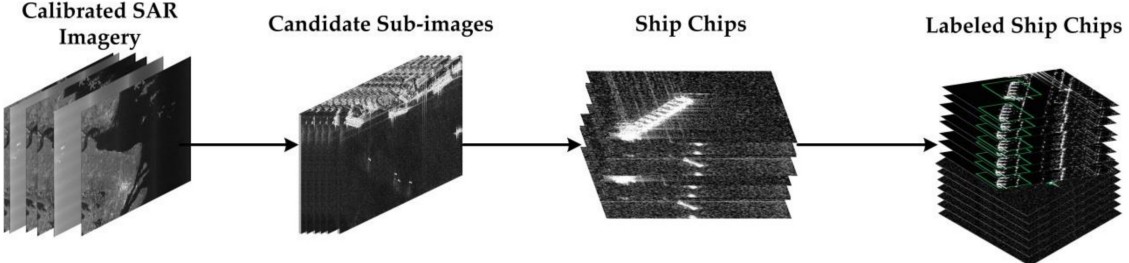

**Figure 5.** The procedures for building the labeled dataset.

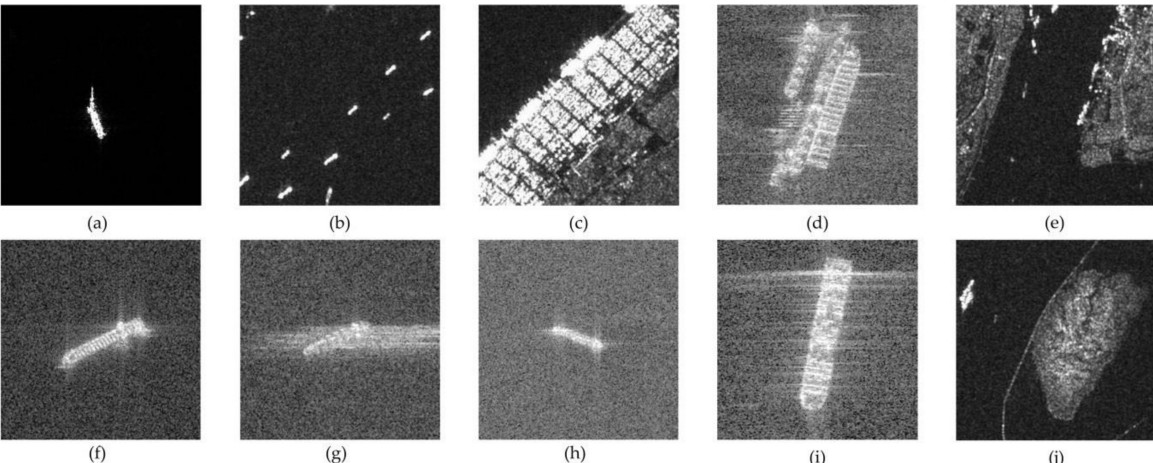

**Figure 6.** Some examples of ship chips. (**a**,**h**) are the cropped ship chips from the Gaofen-3 images with FS1 imaging mode. (**b**,**c**,**e**,**j**) are the cropped ship chips from the Sentinel-1 images with IW imaging mode. (**d**,**f**,**g**,**i**) are the cropped ship chips from the Gaofen-3 images with UFS imaging mode.

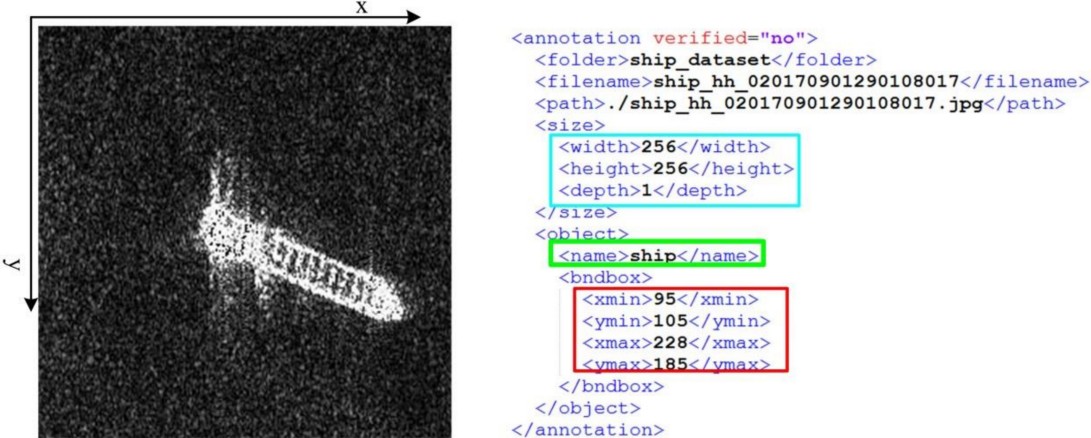

**Figure 7.** Detailed information of a labeled ship chip. The red, green, and cyan rectangles indicate the ship location, the object name, and the ship chip shape in the image on the left.

## 2.3. Properties of the Dataset

There are two main factors for SAR ship detection with deep learning, including ship size and background influencing detection performance. Compared with those objects in the optical dataset, the ship size is relatively quite small. Therefore, the ship may be a pixel and miss the characteristics of ships on a feature map after subsampling several times in deep convolutional networks (ConvNets), thus making it difficult to distinguish ships. Since the targets on the backgrounds of the ships, such as a building, harbor, noise, and islands may share similar backscattering mechanisms to ships, thus leading to false alarms [16,20,31].

### 2.3.1. Multi-Scale Ship Size

There are three reasons for multiscale ship size. First, the shapes of ships may be inherently different, so ships in the same image have multiple scales. Second, the same ship in various resolution images has different sizes. Finally, as stated in Section 2.1, the SAR images are influenced by its imaging mechanisms. The geometry deformation caused by incidence angle also influence the sizes of ships in SAR images. For example, the ships in Figure 6a,b,f,i have different scales. Since there is no Automatic Identification System (AIS) information for these ship chips, one cannot obtain the exact sizes of the ships. To solve this problem in a manner similar to [32], relative bounding box size, calculated by

$\sqrt{w_{bbox} \times h_{bbox}} / \sqrt{w_{img} \times w_{img}}$, is used to measure the ship size to the ship chip, as shown in Figure 8. Here, $w_{bbox}$ and $h_{bbox}$ are the width and height of the ship bounding box, while $w_{img}$ and $w_{img}$ are the width and height of the ship chip (i.e., 256). Compared with the ship chip, the larger the ship, the higher the relative bounding box size. It is apparent that the ship shapes are multiscale and the ship size is relatively small (less than 0.2).

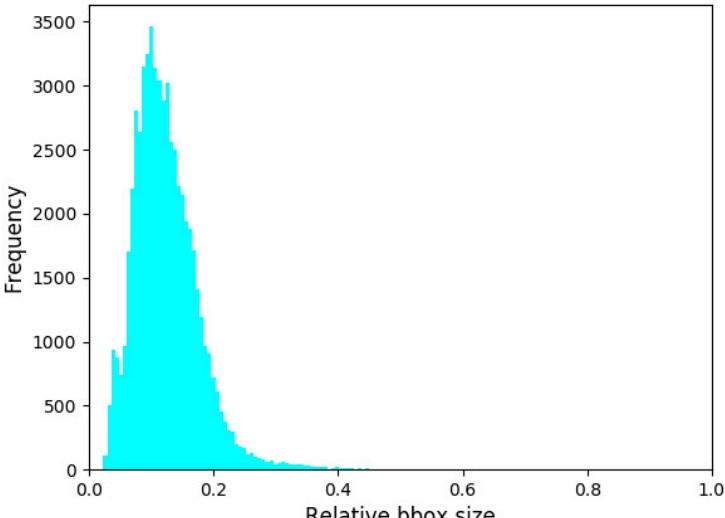

**Figure 8.** The relative bounding-box size distribution of the entire dataset. It is obvious that most ships are of relatively small size (less than 0.2).

### 2.3.2. Complex Backgrounds for Ships

When buildings, islands, or harbors have double backscattering reflections, they may share the similar sigma0 values as ships, leading to many false positives. Therefore, land-ocean segmentation is often used in traditional methods. The deficiency of this method is that land-segmentation limits the speed of ship detection and impedes automatic, end-to-end ship detection. To offset this, the ship chips include ships in complex environments, such as in Figure 6c–e,j. The backgrounds for these four ship chips are harbor, harbor, inshore waters, and island, respectively. Also, the sea conditions may also cause bad conditions for ships. If the ocean is calm, its scattering mechanism is a single reflection as shown in Figure 2. Otherwise, there may be volume scattering that weakens the contrast between ocean and ship. Considering the advantages of learning complex hierarchical features of targets, object detectors in deep learning will be adapted for ship detection and can act as baselines.

## 3. Experimental Results

### 3.1. Related Object Detectors

Currently, there are many object detectors in deep learning for computer vision, such as R-CNN [33], Faster R-CNN [33], feature pyramid networks (FPN) [34], SSD [27], YOLO [35,36], and RetinaNet [37]. These models can be divided into two-stage and one-stage models. Faster R-CNN is the top two-stage performer, and RetinaNet has the best performance among one-stage models. These two models are the baseline for this dataset. Also, since SSD-512 is the top PASCAL VOC dataset [30], it was also chosen as a baseline. Since SSD, Faster R-CNN, and RetinaNet can share the same backbone network to extract features for further object classification and bounding-box regression, one kind of backbone networks will be chosen. Here, the convolutional neural networks VGG [37] named by Visual Geometry Group in the university of Oxford will be selected as the backbone and others, such as residual networks [38] and the Inception network [39–41] will be investigated in future research work. There are two kinds of VGG, i.e., VGG16 and VGG19. Since VGG16 has slighter higher performance

than VGG19, it will be selected as the backbone in this paper. Therefore, VGG16 will be introduced first, followed by SSD, Faster R-CNN, and RetinaNet.

### 3.1.1. VGG16

The architecture of VGG16 [37] has two components, as illustrated in green and brown in Figure 9a: (1) the block-stacked convolutional layer (Conv), the rectified linear unit layer (ReLU), and the pooling layer (Pool) in green; and (2) the fully connected layer (FC) in brown. The first component has five blocks ($C_1$, $C_2$, $C_3$, $C_4$, and $C_5$). Each Conv layer has a kernel size of 3, a stride of 1, and a padding of 1. For each Pool layer, the stride and kernel size are both 2. These blocks are used to distill image features from a low level to a high level. Specifically, the feature map sizes decreased to the half the size of the previous layer, as indicated by the red "0.5×" in Figure 9d. The sizes of the feature maps are only half of that in previous layer after Pool. There are four blocks in the second component, including three FC layers and a softmax to output the probabilities of classes to which the object belongs.

### 3.1.2. Faster R-CNN

Faster R-CNN [26] consists of three main networks: the ConvNets to extract feature maps, the region proposal network (RPN) for generating region proposals, and a network using these proposals for object classification and bounding-box regression [26], as shown in Figure 9b. First, an image is fed to the deep convolutional networks (ConvNets) to obtain its feature maps, and then the RPN is employed to acquire a set of the rectangular object proposal and its corresponding object score. After that, region of interest (ROI) pooling resizes the rectangular object proposals to the same shapes. Finally, these converted object proposals are passed to the classifier to output their corresponding classes and the bounding boxes that include the ships. VGG16 was chosen as the ConvNets for ship detection. The architecture of the Faster R-CNN is illustrated in Figure 9b.

### 3.1.3. SSD

The SSD [27] consists of two parts, including the base network to extract features and the detection part, followed by non-maximum suppression (NMS) obtain the final detection results as shown in Figure 9c. There are many variations for the base network, such as VGG, ResNet, and Inception. Here, VGG is selected. For the detection component, convolutional layer instead of FC layer to acquire the object category and the four indexes of the bounding boxes. It is apparent that SSD uses multi-scale features to detect objects. There are two kinds, including SSD-300 and SSD-512 widely used for object detection. The main difference between them is the size of input, leading to various numbers in default boxes during detections. Particularly, the input sizes for SSD-300 and SSD-512 are 300 and 512, respectively.

### 3.1.4. RetinaNet

The architecture of RetinaNet [42] contains three components: a backbone network and two subnetworks (one for classification and the other for box regression [20]) as shown in Figure 9e. FPN is used to distill multiscale features. FPN [34] has two pathways—a bottom-up pathway and top-down pathway, as shown in Figure 9d. The former usually consists of ConvNets such as VGG16, ResNet, and Inception, and employs them to extract hierarchical features. The latter constructs the high-resolution, multiscale layers from the top layer in the bottom-up pathway. For the top-down, $1 \times 1$ convolution is adapted to reduce the number of feature map channels to 256, and skip-connection is utilized for lateral connections between the corresponding feature maps and the reconstructed layers, which enables the object detector to better predict the location. It acts as a feature extractor with the consideration of the low-level high resolution and high-level low-resolution semantic meaning [38]. Specifically, RetinaNet uses a backbone network such as ResNet, VGG, Inception, or DenseNet [43] to extract higher semantic feature maps, and then FPN is applied to extract the same dimension features with various scales.

After that, these pyramidal features are fed to the two subnets to classify and locate objects as shown in Figure 9e.

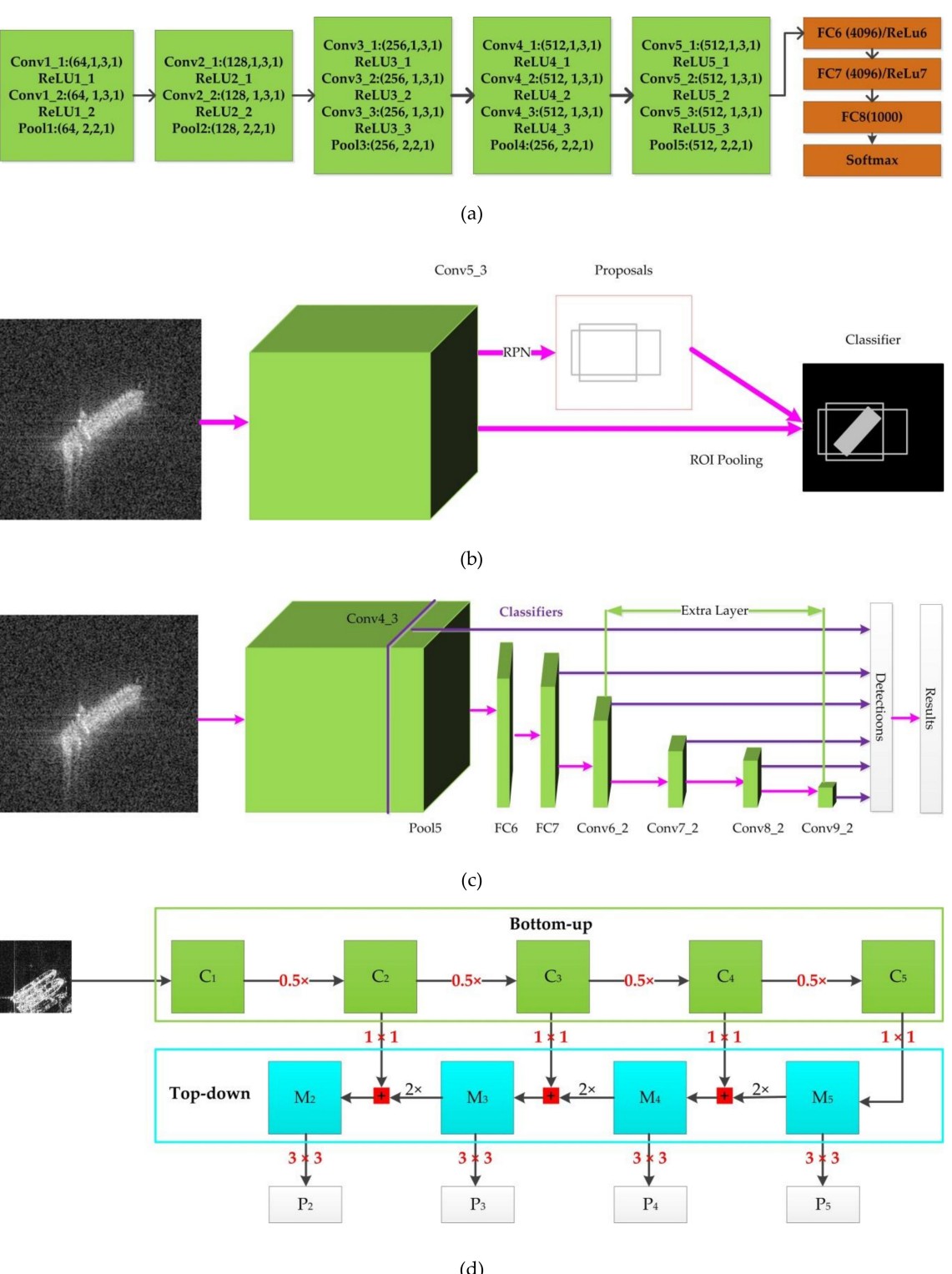

**Figure 9.** *Cont.*

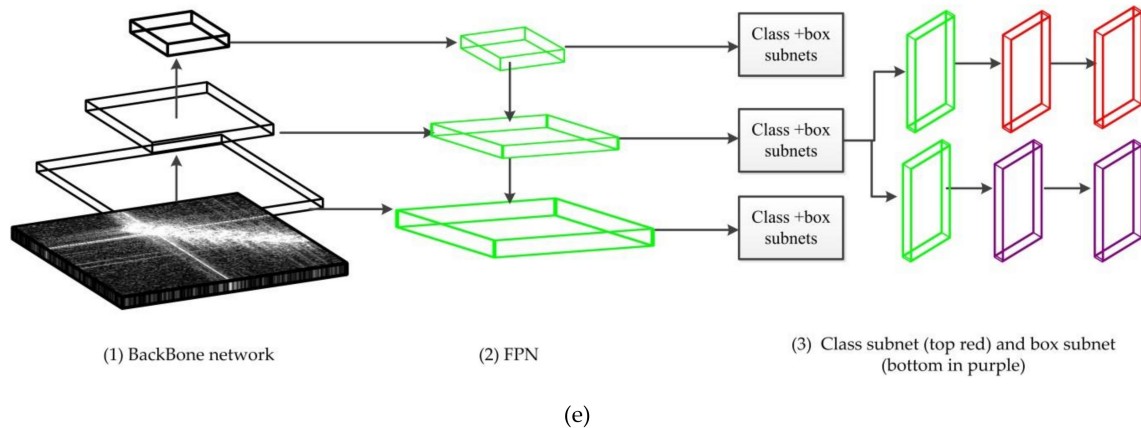

(e)

**Figure 9.** The architecture of Visual Geometry Group (VGG16) (**a**), Faster regions with convolutional neural networks (R-CNN) (**b**), single shot multiBox detector (SSD) (**c**), FPN (**d**), and RetinaNet (**e**).

### 3.2. Training Details

Our experiments were conducted on the Ubuntu 14.04 operating system with an 8GB memory NVIDIA GTX 1070 GPU. The platforms for SSD, Faster R-CNN, and RetinaNet are Caffe [44], TensorFlow [44], and Keras [44], respectively. Considering the SAR characteristics, data augmentation is discarded in our experiments. This is based on the assumption the ship chips after data augmentation may not fit with the SAR imaging mechanisms. For example, horizontal flip cannot satisfy the corresponding geometry deformation. For SSD, there are two types (SSD-300 and SSD-512). During experiments, they share the same learning rate (0.000001) and moment (0.99), whereas they have different batch size (18 for SSD-300 and 2 for SSD-512) due to the GPU's limited memory. Other hypermeters for SSD-300 and SSD-512 are set to be the same as in [27]. For Faster R-CNN, the empirical values for the learning rate, batch size, the moment, and the momentum are 0.0001, 18, 0.99, and 0.0001, respectively. Other hypermeters are set to be the same as in [26]. For RetinaNet, Adam optimization is used with a learning rate of 0.00001 and a batch size of 2. Other hyperparameters are set to be the same as in [42]. In addition, early stopping [21] is used to terminate the training.

### 3.3. Experimental Results and Analysis

### 3.3.1. Experimental Results for Baselines

The results of the four models are shown in Table 2. It is obvious that Faster R-CNN and RetinaNet achieve the worst and the best mean average precision (mAP), respectively. This may be because RetinaNet exploits multiscale features for ship detection, whereas Faster R-CNN does not. Since the feature map size will shrink to half of the previous layer after Pool, the feature map sizes on Conv7_2, Conv8_2, and Conv9_2 will be 1/64, 1/128 and 1/256 of the input image sizes, respectively. Considering that the bounding-box sizes that include the ships are relatively small (as shown in Figure 5), these three layers are removed to construct the new ship detection model for SSD-300 and SSD-512. The modified SSD is shown in Figure 10. It is clear that the number of the parameters in SSD will be decreased. From Table 2, it is obvious that the performance of both modified SSD-300 and SSD-512 remain almost the same and require less training time, demonstrating the benefits of SAR characteristics when designing the ship detection models.

**Table 2.** Ship detection mean average precision (mAP) of four models.

| Model | Input Size (pixels) | mAP (%) | Training Time (minutes) |
|---|---|---|---|
| SSD-300 | $300 \times 300$ | 88.32 | 193.25 |
| SSD-512 | $512 \times 512$ | 89.43 | 253.01 |
| Faster R-CNN | $600 \times 800$ | 88.26 | 531.4 |
| RetinaNet | $800 \times 800$ | 91.36 | 650.77 |
| Modified SSD-300 | $300 \times 300$ | 88.26 | 127.89 |
| Modified SSD-512 | $512 \times 512$ | 89.07 | 221.83 |

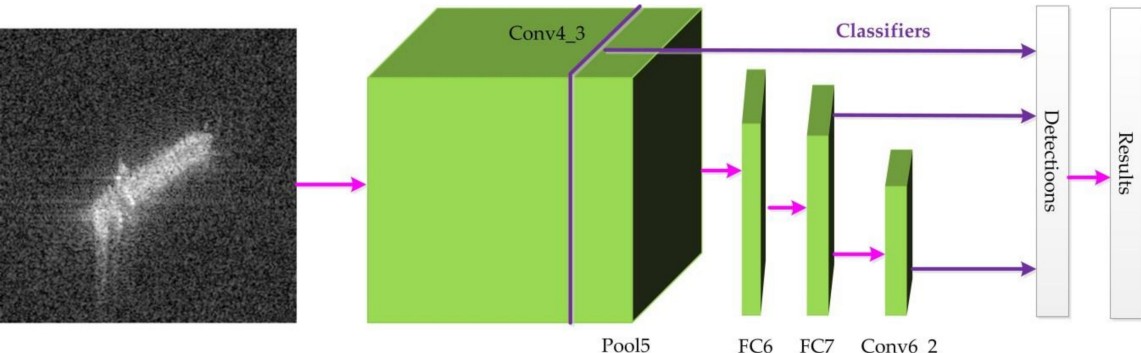

**Figure 10.** The architecture of modified SSD, which removes the Conv7_2, Conv8_2, and Conv9_2 from SSD in Figure 9c.

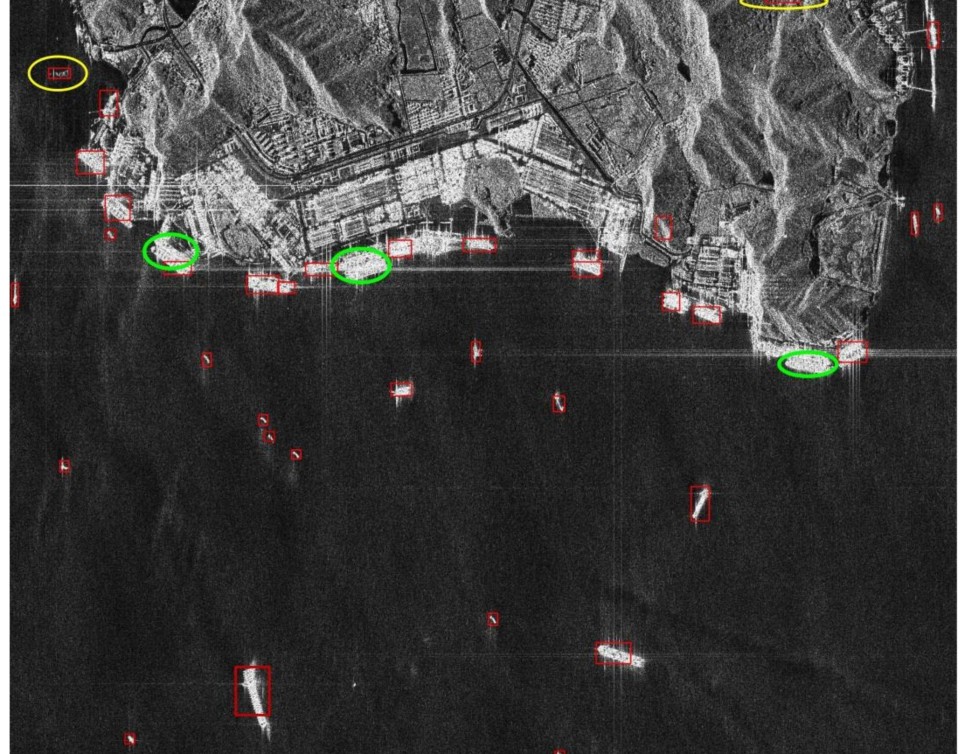

**Figure 11.** Ship detection results under complex backgrounds. Red rectangles, green ellipses, and yellow ellipses indicate the detected ships, missing ships, and false alarms, respectively.

### 3.3.2. Experimental Results for Generalization

As stated in Section 2.3.2, the backgrounds of ship chips are complex as shown in Figure 6. The trained models achieve promising mAP on the test dataset. To further validate whether the trained models can be used to detect ships with complex backgrounds on a new SAR imagery, one Chinese

Gaofen-3 image is used. Especially, modified SSD-300 with the lowest mAP is selected. The detection results are shown in Figure 11. It is obvious that modified SSD-300 can detect almost all the multiscale ships, except those that are side by side in the harbor. This is because modified SSD-300 employs multiscale features to detect ships, as shown in Figure 10. Moreover, even if there are buildings with high sigma0 values, they are not false positives, demonstrating the benefits of this dataset for training a model to detect ships without land-ocean segmentation.

## 4. Discussion

Many factors, such as incidence angle, ship size, wind speed, and metocean parameters influence the ship detectability [45,46]. Typically, it is easy to detect ships in high incidence angle, low wind speed, or low sea state [45]. Here the incidence angle, wind speed, and metocean parameters mainly influence the sigma0 values contrast between the ocean and the ship, thus making the background of ships complex. Especially, the fast wind speed and bad metocean induce turbulent water, which may have volume scattering, thus leading to quite complicated surroundings of ships. Therefore, ship size and background are focused in this paper. Future work will be conducted regarding the evaluation of ship detectability on various specific conditions.

Ships with complex backgrounds are multiscale and are relatively small. In this paper, we provide a large-volume dataset for ship detection with SAR images. Since AIS information is not available, some ships in the dataset labeled by SAR experts may be incorrect, to some extent. We believe this dataset can still promote the development of object detectors just like ImageNet [22]. Moreover, the correction of this dataset will be ongoing. If possible, the AIS information will be added.

Section 3.3.2 makes it clear that the modified SSD-300 cannot detect ships that are side by side in the harbor, as shown by green ellipses in Figure 10. This may be because of the lack of ship chips in the training dataset. To make this dataset more powerful, this kind of ship chip will be added to our dataset.

As for the baselines, we only conducted the experiments based on state-of-the-art object detectors. Currently, there is some computer-vision work related to detecting small objects [47–50]. The core ingredient is the multiscale feature with high resolution to improve detection accuracy [47,48]. Considering the relatively small ships, these models can be adapted for ship detection based on our dataset. One can also design new ship detection approaches using our dataset.

## 5. Conclusions

In this study, a large-volume dataset with 43,819 ship chips was constructed to boost the application of object detectors for SAR ship detection. The ships in this dataset have numerous annotations mainly varying in scale and background and are the first to tackle ship detection with complex surroundings. To build the baselines, some state-of-the-art models were evaluated. Moreover, considering the relatively small sizes of the ships, modified SSD was also proposed as a baseline for ship detection. The experiment results revealed that (1) although there is no land-ocean segmentation, the models trained on these data can detect multiscale ships with low numbers of false positives under complex backgrounds and (2) considering the characteristics of ships in SAR imagery, modified object detectors (Modified SSD-300 and Modified SSD-512) can achieve comparable results. We believe this dataset will improve ship detection without land-ocean segmentation, leading to a new framework for ship detection with object detectors in deep learning.

**Author Contributions:** Y.W. was mainly responsible for the construction of ship detection dataset, conceived the manuscript, and conducted the experiments. C.W. supervised the experiments. H.Z. helped acquire the Chinese Gaofen-3 SAR images and also contributed to the organization of the paper and the experimental analysis. S.W. and Y.D. were also responsible for the dataset.

**Funding:** This research was funded by National Key Research and Development Program of China (2016YFB0501501) and the National Natural Science Foundation of China under Grant 41331176.

**Acknowledgments:** We own many thanks to China Center for Resources Satellite Data and Application and Copernicus Open Access Hub for providing Gaofen-3 images and Sentinel-1 images, respectively.

**Conflicts of Interest:** The authors declare no conflict of interest.

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
