# Peer review of "A SAR Dataset of Ship Detection for Deep Learning under Complex Backgrounds"

_remotesensing, doi:10.3390/rs11070765_

Round 1

Reviewer 1 Report

Authors investigated ship detection using deep learning approaches trained on SAR dataset collected in complex real-world backgrounds. Space-borne satellites are launched leading to more synthetic aperture radar (SAR) images being available than ever. This data can help in dynamic ship monitoring. There are several opensource public dataset that are leveraged for training deep learning models for high accuracy object detectors. Due to the unavailability of massive labeled datasets, object detectors for SAR ship detection are still in infancy. Therefore, authors established a SAR dataset labeled by experts for deep learning-based object detectors. It contains 102 Chinese Gaofen-3 images and 108 Sentinel-1 images. A total of 43,819 ship chips with 256 pixels in both range and azimuth are included in this dataset where these ships have distinct scales and backgrounds. Authors modified the state-of-the-art object detectors trained using natural images as baselines. Comparative experiments with object detectors achieve high mean average precision (mAP) on the dataset. It also showed good generalization performance on new SAR imagery without land-ocean segmentation. Clearly, the dataset developed by the authors have several benefits. This dataset has annotations with variations in scale and background facilitating ship detection in complex surroundings.

Author Response

Dear Section Managing Editor,                                                                                       10 March 2019Dear reviewers,

Remote Sensing

Manuscript ID remotesensing-454244 entitled "A SAR Dataset of Ship Detection for Deep Learning under Complex Backgrounds".

We appreciate the thorough reviews provided by the referees. We agree with these suggestions and have revised the manuscript accordingly. Below is our response to their comments resulting in a number of clarifications. We hope these revisions resolve the problems and uncertainties pointed out by the referees. In the manuscript and this file, the red, magenta, and blue parts are revisions suggested by the three referees, respectively. The underline parts in the manuscript are those changed contents to improve the expressions.

Regards,

Hong Zhang

zhanghong@radi.ac.cn

Reviewer 2 Report

A SAR Dataset of Ship Detection for Deep Learning under Complex Backgrounds:  

A SAR ship detection dataset under complex backgrounds (like harbor, inshore waters, and island) is constructed. Modified state-of-the-art object detectors, including Faster RCNN, SSD, and RetinaNet are adapted to ship detection and can be baselines. Faster RCNN is the top two-stage performer, and RetinaNet has the best performance among one-stage models.

Authors explained two main factors for SAR ship detection with deep learning, including ship size and background influencing detection performance. They achieved the speed of ship detection and automatic, end-to-end ship detection. Authors worked on a large-volume dataset for ship detection with SAR images. the models trained on these data can detect multiscale ships with low numbers of false positives under complex backgrounds.

Good and interesting work.

Author Response

Dear Section Managing Editor,

Dear the referees                                                                                                  10 March 2019

Remote Sensing

Manuscript ID remotesensing-454244 entitled "A SAR Dataset of Ship Detection for Deep Learning under Complex Backgrounds".

We appreciate the thorough reviews provided by the referees. We agree with these suggestions and have revised the manuscript accordingly. Below is our response to their comments resulting in a number of clarifications. We hope these revisions resolve the problems and uncertainties pointed out by the referees. In the manuscript and this file, the red, magenta, and blue parts are revisions suggested by the three referees, respectively. The underline parts in the manuscript are those changed contents to improve the expressions.

Regards,

Hong Zhang

zhanghong@radi.ac.cn

Reviewer 3 Report

some editorial changes are required. for example:

lines 127-129 are not easy to understand the sentence, rewrite it.

Please add the definition of sigma0 or the reference for it when it is used first.

You mentioned that the water has a significant effect and it makes it difficult to detect the ships when it is not calm. Can it be taken care of in your model? if not what would be the effect and is there a further investigation on that? if yes is there any comparisons between yours and other models?

Have you tried to detect ships in more turbulent water?

Can you compare the result of your modeling and detection with another state of the art classification to show the improvement?

Author Response

Dear Section Managing Editor,

Dear the reviewers                                                                                                  10 March 2019

Remote Sensing

Manuscript ID remotesensing-454244 entitled "A SAR Dataset of Ship Detection for Deep Learning under Complex Backgrounds".

We appreciate the thorough reviews provided by the referees. We agree with these suggestions and have revised the manuscript accordingly. Below is our response to their comments resulting in a number of clarifications. We hope these revisions resolve the problems and uncertainties pointed out by the referees. In the manuscript and this file, the red, magenta, and blue parts are revisions suggested by the three referees, respectively. The underline parts in the manuscript are those changed contents to improve the expressions.

Regards,

Hong Zhang

zhanghong@radi.ac.cn
